

# Gut content metabarcoding of specialized feeders is not a replacement for environmental DNA assays of seawater in reef environments

Joseph D. DiBattista[1,2,*], Shang Yin Vanson Liu[3,*],
Maarten De Brauwer[4], Shaun P. Wilkinson[5], Katrina West[6],
Adam Koziol[1,7] and Michael Bunce[8]

[1] School of Molecular and Life Sciences, Curtin University, Perth, WA, Australia
[2] Australian Museum Research Institute, Australian Museum, Sydney, NSW, Australia
[3] Department of Marine Biotechnology and Resources, National Sun Yat-Sen University, Kaohsiung, Taiwan
[4] CSIRO Oceans and Atmosphere, Hobart, TAS, Australia
[5] School of Biological Sciences, Victoria University of Wellington, Wellington, New Zealand
[6] CSIRO Australian National Fish Collection, CSIRO, Hobart, TAS, Australia
[7] Department of Clinical Medicine, Aalborg University, Copenhagen, Denmark
[8] Institute of Environmental Science and Research, Kenepuru, Porirua, New Zealand
* These authors contributed equally to this work.

Corresponding author
Shang Yin Vanson Liu,
oceandiver6426@gmail.com

## ABSTRACT

In tropical marine ecosystems, the coral-based diet of benthic-feeding reef fishes provides a window into the composition and health of coral reefs. In this study, for the first time, we compare multi-assay metabarcoding sequences of environmental DNA (eDNA) isolated from seawater and partially digested gut items from an obligate corallivore butterflyfish (*Chaetodon lunulatus*) resident to coral reef sites in the South China Sea. We specifically tested the proportional and statistical overlap of the different approaches (seawater *vs* gut content metabarcoding) in characterizing eukaryotic community composition on coral reefs. Based on 18S and ITS2 sequence data, which differed in their taxonomic sensitivity, we found that gut content detections were only partially representative of the eukaryotic communities detected in the seawater based on low levels of taxonomic overlap (3 to 21%) and significant differences between the sampling approaches. Overall, our results indicate that dietary metabarcoding of specialized feeders can be complimentary to, but is no replacement for, more comprehensive environmental DNA assays of reef environments that might include the processing of different substrates (seawater, sediment, plankton) or traditional observational surveys. These molecular assays, in tandem, might be best suited to highly productive but cryptic oceanic environments (kelp forests, seagrass meadows) that contain an abundance of organisms that are often small, epiphytic, symbiotic, or cryptic.

## INTRODUCTION

An important component of isolated islands and atolls in both tropical and subtropical environments are zooxanthellate scleractinian corals, a matrix of calcium carbonate skeleton and live animal polyps that provide habitat, food, and refuge. This is particularly true for the approximately 128 species of fish that feed from live corals (corallivores), which are often associated with healthy coral reef environments (*Cole, Pratchett & Jones, 2008*). The cover of preferred (or healthy) corals can be a primary determinant of corallivore abundance and physiological condition (*Pratchett et al., 2004*; *Berumen, Pratchett & McCormick, 2005*; *Pratchett & Berumen, 2008*).

Corallivorous fish often show distinct prey preferences for coral type, but these are often hard to identify and/or quantify both *in situ* and within dissected stomach samples. Furthermore, post-processing techniques (*i.e.*, the morphological analysis of stomach samples) are biased by prey decomposition rates and homogenization, as well as the challenges of visually identifying soft-bodied organisms (*Nagelkerken et al., 2009*). To overcome challenges associated with traditional morphological diet classification, the use of DNA metabarcoding has been increasingly applied to analyse and classify trace DNA fragments of prey extracted from stomach, faecal, and regurgitate samples of tropical fishes (*Leray et al., 2013*; *Leray, Meyer & Mills, 2015*; *Leray et al., 2019*; *Devloo-Delva et al., 2019*; *Casey et al., 2019*; *Coker et al., 2023*). Whilst this powerful technique largely circumvents the need for morphological taxonomic expertise, it should be noted that it can still be biased by progressive prey decomposition and DNA degradation (*Carreon-Martinez et al., 2011*), as well as the inability to distinguish between targeted prey and incidental ingestion.

High-throughput next-generation sequencing (NGS) in combination with DNA metabarcoding using universal PCR assays across DNA regions such as the 18S small subunit ribosomal rRNA (hereafter 18S) or internal transcribed spacer 2 (hereafter ITS2), can greatly increase the range of items detected in stomach samples (for 18S; see *Coker et al., 2023*) or resolution of basal metazoans identified (for ITS2; see *Alexander et al., 2020*; *West et al., 2022*). In a similar approach to diet metabarcoding, environmental DNA (eDNA) profiles obtained from seawater, sediment, or additional substrates can also provide a broader picture of the biological assemblages present at a particular site (*Kelly et al., 2017*; *Boussarie et al., 2018*; *Koziol et al., 2019*; *Stat et al., 2017, 2019*; *DiBattista et al., 2019, 2020*; *West et al., 2020*), but rarely has this approach been compared with diet metabarcoding. Indeed, in the tropical marine environment, there are examples of fish diet linked to benthic quadrat data (*e.g.*, herbivorous diet *vs* environmental algal composition; *Stamoulis et al., 2017*), but there are fewer comparisons, if any, of DNA presence/absence data generated from seawater and the diet of fish feeding concurrently at the same site (noting that some of the surrounding seawater will inevitably be ingested alongside its targeted prey).

In this metabarcoding study we explore the diet of one species of butterflyfish in the family Chaetodontidae, which contain the highest proportion of corallivores (~54%) compared to any other family of fish (*Cole & Pratchett, 2014*). We also collected seawater

samples to rapidly characterise benthic composition in areas surrounding butterflyfish feeding points. Our focal species was *Chaetodon lunulatus*, which is considered an obligate hard corallivore (*Pratchett, 2005*, *2007*; *Nagelkerken et al., 2009*) but may feed more broadly on benthic invertebrates.

We chose to focus our eDNA and gut content sampling at Dongsha Atoll, an island 415 km south of Taiwan in the South China Sea because it hosts undamaged corals and corallivore butterflyfish. Dongsha Atoll has been identified as a fast-acidified ocean site and has both independent coral reef and seagrass ecosystems that harbor high biodiversity (*DeCarlo et al., 2015*). The atoll is comprised of three reef components–the fore reef (including reef crest), back reef, and lagoon–with the lagoon limited in open ocean water exchange and experiencing higher summer temperatures on average. We therefore might expect a higher level of seawater retention in the lagoon *vs* predominantly fringing or coastal coral reef ecosystems. Moreover, the corals on the fore reef were not affected by the 1998 worldwide bleaching event, whereas the lagoon corals experienced almost complete mortality at the same time (*Dai, 2004*), selectively killing almost all acroporids and pocilloporids (*Fang, 1998*; *Li et al., 2000*). In contrast, the coral cover at the northern and eastern outer reef slopes of the atoll remained high and virtually undamaged during this same warming event. More recently, corals in the lagoon were observed bleaching in the summer of 2015 when water temperatures became too high (*i.e.*, over 32 °C for a sustained period; *DeCarlo et al., 2017a*). That said, most symbiotic zooxanthellae had returned to the corals when observed in the spring of 2016 along with residual coral mortality in some areas of the atoll. Dongsha Atoll therefore provides the ideal setting to assess how closely butterflyfish gut content and eDNA ally among different types and quality of coral reef habitat.

In this study, using eDNA metabarcoding with a universal 18S assay that broadly targets metazoans, as well as an ITS2 assay that targets largely cnidarians and sponges, we tested whether our focal butterflyfish species (an obligate corallivore) feeds more broadly on benthic invertebrates, as well as the effectiveness of the different sampling techniques (seawater *vs* gut content) in characterizing eukaryotic community composition on coral reefs. In short, do these two approaches display similar biodiversity profiles and what is their utility in tracking future indicators of change on coral reefs?

## MATERIALS AND METHODS

### Sample collection

Sampling was conducted over 7 days in April and May 2018 at 10 sites around Dongsha Atoll within the lagoon, outside the lagoon, and on the seagrass beds (Fig. 1). No more than two sampling sites were visited on any given day (Appendix S4). At sampling sites where seawater was collected, six 1 L replicates were obtained from a dive boat adjacent to the relevant habitat (at or within 2 m horizontal distance from the coral reef) and 30 cm below the surface using sterile Nalgene bottles ($N = 36$ in total). All water samples were immediately stored on ice in an enclosed cooler and filtered at the Dongsha Atoll Research Station (DARS). Each sample was individually filtered across Pall 0.2 μm Supor[®] polyethersulfone membranes using a Pall Sentino[®] Microbiology pump (Pall Corporation,

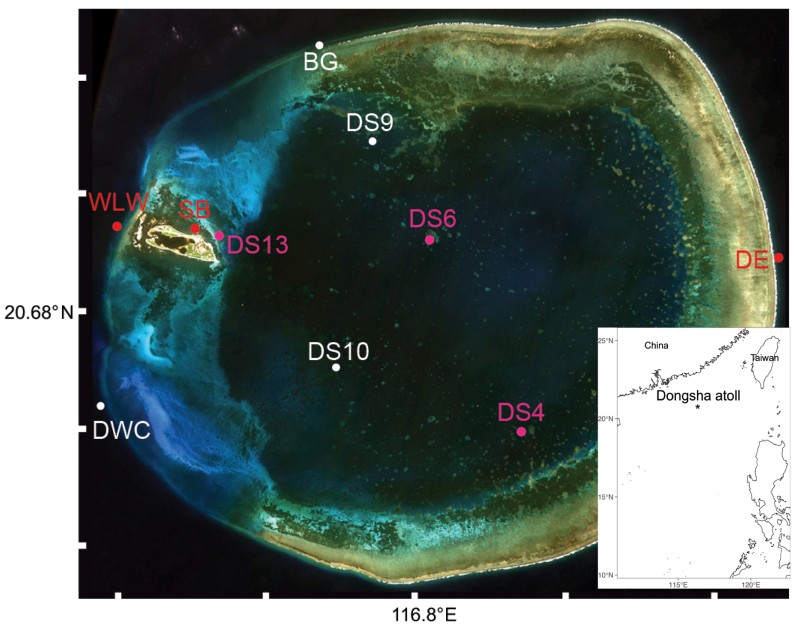

**Figure 1 Sampling sites (N = 10) of seawater and oval butterflyfish (*Chaetodon lunulatus*) gut contents at Dongsha Atoll (Taiwan) in the South China Sea.** Red dots indicate sites where only seawater was sampled, white dots indicate sites where butterflyfish gut contents only were sampled, and pink dots indicate sites where both were sampled. Sample site abbreviations are as follows: Dongsha Site 4 = DS4; Dongsha Site 6 = DS6; Dongsha Site 13 = DS13; Dongsha East = DE; West Lagoon Wreck = WLW; Northeast Seagrass Bed = NSB; Dongsha West Channel = DWC; Dongsha Site 9 = DS9; Dongsha Site 10 = DS10; Boo's Groove (N Logger) = BG. The bottom layer was generated by the ggOceanMaps and ggplot2 packages in RStudio (*R Core Team, 2015*) and the satellite image of Dongsha Atoll was sourced from the Taiwanese government under the following Open Government Data License, v 1.0 (https://data.gov.tw/en/license).

Port Washington, NY, USA) within 6 h of collection. The filtration apparatus was cleaned by soaking in 10% bleach between samples for at least 15 min. We favoured a 1 L filtration volume as previous research has shown that this provides an acceptable compromise between the time to process samples (10 to 15 min per filter replicate) and the biodiversity uncovered in shallow coral reef environments (*e.g.*, *West et al., 2020*; but also see *Mathon et al., 2022*). Filters were stored at −20 °C until processing in the Trace and Environmental DNA (TrEnD) Laboratory at Curtin University in Perth, Australia.

Adult specimens of a single species of butterflyfish (*Chaetodon lunulatus*), a known corallivore, were collected using pole spear on SCUBA taking care not to impale the stomach cavity, which would contaminate the gut contents (N = 40; Appendix S4). Fish collections were undertaken in accordance with the policies and procedures drafted by the Animal Care and Use Committee of National Sun Yat-Sen University. Most fish were dispatched by the action of the spear, but those that were not, were subject to immediate ike jime and placed in an ice slurry on board the vessel. Field permissions to undertake the research were obtained from the Ministry of Interior in Taiwan (Marine National Park Headquarters) under permit No. 1070001035. Following collection, fish were kept on ice in individual Ziploc® bags until storing them at −20°C to mitigate further DNA degradation. To remove gut contents, the fish were partially thawed, the gut cavity was opened, and the
entire semi-frozen stomach contents were placed into 2 ml sample tubes with sterile 80% ethanol taking care not to include the stomach lining; these were immediately stored at −20°C until processing in the Benthic Organisms and Molecular Ecology Lab at National Sun Yat-Sen University. Between each sample, all tools and trays were either replaced or rinsed in bleach (10%) and sterile ethanol (99%).

Due to the patchy distribution of *Chaetodon lunulatus* across Dongsha Atoll, fish were not sampled at all sites where seawater was initially collected. That said, the distributions of collected fishes from a single sampling site were unlikely to overlap with adjacent sampling sites given that the home range of this species is small (average = 117 m$^2$; range = 2.3–976.3 m$^2$; *Cowlishaw, 2014*).

### DNA extraction

Following *DiBattista et al. (2020)*, DNA bound to filter membranes was extracted using Qiagen DNeasy Blood and Tissue kits (Qiagen, Hilden, Germany) that was automated on a Qiacube (Qiagen, Hilden, Germany) to minimise human handling and cross-contamination in the TrEnD Laboratory with the following modifications to the standard protocol: 540 µl of ATL lysis buffer and 60 µl of Proteinase K. DNA extraction controls (blanks) were carried out for every set of 12 extractions. Half of each filter paper was manually shredded with sterilised dissecting scissors prior to immersion in the extraction buffer; the other half of the filter paper was re-frozen at −80 °C to serve as a permanent voucher.

Between 58 and 260 mg of gut contents were spooned into 2 or 7 ml Minilys homogenizing vials depending on the volume of the sample. Samples were homogenized manually by using a micropestle (Geneaid Biotech Ltd, Taiwan, China) for 30 s and then placed in a sterile 1.5 ml tube. Due to the co-purification of inhibitors in stomach samples, DNA extractions were performed using the QIAamp DNA stool kit (Qiagen, Hilden, Germany) according to the standard protocol in a laminar flow hood. DNA extraction controls, for which all steps remained the same except for the addition of stomach contents, were included.

### Fusion-tag qPCR

Following *DiBattista et al. (2020)*, a primer set targeting 18S (V1-3 hypervariable region; 18S_uni_1F: 5′–GCCAGTAGTCATATGCTTGTCT–3′; 18S_uni_400R: 5′–GCCTGCTGCCTTCCTT–3′; *Pochon et al., 2013*) was used to maximise detection of general eukaryotes, hereafter the 18Suni assay. We used a second assay targeting ITS2 (scl58SF: 5′–GARTCTTTGAACGCAAATGGC–3′; scl28SR: 5′–GCTTATTAATATGCTTAAATTCAGCG–3′; *Brian, Davy & Wilkinson, 2019*) to increase the taxonomic resolution of identified Anthozoa (Phylum Cnidaria) and Demospongiae (Phylum Porifera) within each environmental sample, hereafter the ITS2 assay (also referred to as "CP1 assay" in *Alexander et al. (2020)*).

As per *DiBattista et al. (2020)*, quantitative PCR (qPCR) experiments for both seawater and gut content extracts were set up in a separate ultra-clean laboratory at Curtin University designed for trace DNA work using a QIAgility robotics platform (Qiagen,
Hilden, Germany). For more details on contamination mitigation measures, please see *DiBattista et al. (2019, 2020)*. In brief, fusion-tag qPCR was performed with each extract in triplicate on a StepOnePlus Real-Time PCR System (Applied Biosystems, Foster City, CA, USA) under the following conditions for 18Suni: initial denaturation at 95 °C for 5 min, followed by 45 cycles of 30 s at 95 °C, 30 s at 52 °C, and 45 s at 72 °C, with a final extension for 10 min at 72° C. PCR was performed under the following conditions for ITS2: initial denaturation at 95 °C for 5 min, followed by 45 cycles of 30 s at 95 °C, 30 s at 55 °C, and 45 s at 72 °C, with a final extension for 10 min at 72° C. These primers and PCR conditions have been optimized and applied elsewhere (*Alexander et al., 2020*; *DiBattista et al., 2019*, *2020*, *2022*; *West et al., 2020*). Moreover, the primer design incorporated indexes in both the forward and reverse primer, which allowed us to tag individual PCR replicates of individual samples by a unique combination of tags on the forward and reverse primers. To check for contamination, non-template control (labelled as NTC) PCR reactions were run alongside the template PCR reactions, which only contained master mix including the assay primers.

Libraries for sequencing were made by pooling amplicons into equimolar ratios based on PCR Ct values and the endpoint of amplification curves, which were then size selected using a Pippin Prep (Sage Science, Beverly, MA, USA; 100 to 600 bp) and purified using the Qiaquick PCR Purification Kit (Qiagen Inc, Venlo, The Netherlands). The volume of purified library added to the sequencing run was determined against DNA standards of known molarity on a LabChip GX Touch (PerkinElmer Health Sciences, Waltham, MA, USA). Final libraries were sequenced in a paired end approach using a 500 cycle MiSeq® V2 Reagent Kit and standard flow cell on an Illumina MiSeq platform (Illumina, San Diego, CA, USA) located in the TrEnD Laboratory.

## Bioinformatic filtering

Sequence data were quality filtered (QF) prior to ZOTU (Zero-radius Operational Taxonomic Unit) creation and taxonomic assignment. ZOTUs are equivalent to amplicon sequence variants (ASVs) and provide a more precise measurement of sequence variation (also see *Edgar, 2018*). Metabarcoding reads recovered by paired-end sequencing were first merged using the Illumina MiSeq analysis software under the default settings (*i.e.*, minimum 10 bp overlap between read 1 and read 2). Sequences were then assigned to samples based on their unique index combinations and trimmed in Geneious® Pro v 4.8.4 (*Drummond et al., 2009*). Only those sequences with 100% identity matches to Illumina adaptors, index barcodes, and template specific oligonucleotides were kept for downstream analyses. Sequences were further processed in Geneious by trimming for base quality (phred score > 30) and minimum length (18Suni, 200 base pairs; ITS2, 100 base pairs) using BBDuk v 36.92 (*Bushnell, 2017*). USEARCH v 11.0.667 (*Edgar, 2010*) was then used for dereplication into unique sequences, removing singletons, removing chimeric sequences, and generating a ZOTU (minimum number of sequences for ZOTU creation = 10) table based on the UNOISE algorithm, which performs denoising (error-correction) of the amplicon reads. USEARCH and the taxonomic assignment pipeline outlined below (BLAST, LULU, and LCA assignment) was facilitated by an in-house script

developed by *Mousavi-Derazmahalleh et al. (2021)* implemented on a high-performance computing platform at the Pawsey Supercomputing Centre in Western Australia.

## Taxonomic assignment

ZOTUs were queried against the National Centre for Biotechnology Information's (NCBI) GenBank nucleotide database (accessed in 2020/2021) using BLASTn v2.10.1 with the following conservative settings: percentage identity (99), query coverage of 100, best hit score edge of 0.05, best hit overhang of 0.25, and an E-value of $1e^{-3}$. LULU (*Frøslev et al., 2017*) was then run to curate the assignments and eliminate any remaining redundant sequences, prior to assigning the lowest common ancestor (LCA) with a custom Python script, with the default parameters: minimum_ratio_type = min, minimum_ratio = 1, minimum_match = 84, minimum_relative_cooccurence = 0.95. Based on the LCA process, some ZOTUs were assigned at the species level, some at the genus level, some at the family level, and the remainder only at higher taxonomic ranks, as noted. All additional information related to this bioinformatic workflow is in *Mousavi-Derazmahalleh et al. (2021)* as well as freely available scripts at the following link (https://github.com/mahsa-mousavi/eDNAFlow). All ZOTUs that were detected from extraction controls were removed from further analyses (Appendix S1). This included ZOTUs assigned to *Trichoderma*, Malasseziomycetes, Chlorophyta, Oedogoniaceae, uncultured Chlorophyta, uncultured Chytridiomycota, *Cultellothrix coemeterii*, Eukaryota, Dictyoceratida, *Ephydatia*, *Brassica rapa*, *Blastocystis* sp., and uncultured eukaryotes for the 18Suni assay. This also included ZOTUs assigned to *Montipora*, *Montipora digitata*, *Montipora flabellata*, Poritidae, *Porites*, *Porites lobata*, *Porites lutea*, Eukaryota, and Dysideidae for the ITS2 assay. Note that ZOTU1841 was assigned to a primate sequence and thus also removed from the 18Suni analysis. Also note that "BLANK4" was the only extraction blank where the number of sequences passed the filtering thresholds for the ITS2 analysis. Order level dendrograms for the 18Suni and ITS2 assay taxonomic assignments were constructed using a phylogenetic tree generator based on NCBI taxonomy (phyloT v2022.3 and iTOL v5; *Letunic & Bork, 2021*).

## Statistical analyses

Initial PERMANOVA tests PRIMER v7 (*Clarke & Gorley, 2015*) across all samples based on presence/absence Jaccard similarity matrices showed that the community assemblage recovered with each method (water and gut content ZOTUs) was significantly different (18Suni: df = 1, MS = 68,668, Pseudo-F = 43.426, $p < 0.0001$; ITS2: df = 1, MS = 33,147, Pseudo-F = 11.089, $p < 0.0001$), resulting in a near complete lack of overlap for the detected ZOTUs (Appendix S2). Subsequent analyses further investigating trends across all sites were performed on subsets of the data separated by sampling approach (Appendix S3 for PRIMER input). The site composition of ZOTUs were therefore analysed separately for 18Suni and ITS2 in presence/absence format using PRIMER v7.

To test whether eDNA detections from seawater and butterflyfish gut contents displayed similar biodiversity profiles, we only considered sites where both approaches were applied (DS4, DS6, DS13; also see Appendix S1–S4). For these analyses, Jaccard

similarity matrices were calculated based on presence/absence transformed data. PERMANOVA (9,999 permutations) was subsequently used to quantify these differences and principal coordinate analysis (PCO) was used to visualise the data (*Anderson & Willis, 2003*).

## RESULTS

### Total biodiversity detected

A total of 3,206,449 amplicon reads pertaining to the 18S gene that passed QF were retrieved from 36 seawater samples (average per replicate = 89,068 ± 7,085 SEM) and 3,058,957 amplicon reads were retrieved from 38 butterflyfish gut samples (average per replicate = 80,498 ± 10,995 SEM; Appendix S4). Overall, the 18Suni metabarcoding data produced assignments pertaining to 12 eukaryotic phyla, 22 classes, 40 orders, 43 families, and 38 genera across the combined seawater and butterflyfish gut samples (Appendix S1; Fig. 2A). The average number of ZOTUs assigned per replicate of seawater was 115.94 ± 8.77 SEM and for gut content it was 9.16 ± 0.60 SEM. At sites where both seawater and gut contents were sampled, on average, 93.31 ± 1.60% SEM of the ZOTUs were unique to seawater, 3.39 ± 1.36% SEM of the ZOTUs were unique to gut contents, and 3.30 ± 0.37% SEM of the ZOTUs were shared between the two sampling approaches. Here, one family of stony corals was identified as unique to gut contents (Acroporidae), whereas 34 families were identified as unique to seawater, with the greatest representation included in the classes Bivalvia (marine and freshwater molluscs; five families), Demospongiae (sponges; five families), Dinophyceae (dinoflagellates; four families), Florideophyceae (red algae; three families), and Polychaeta (bristle worms; four families). Two families of stony coral (Agariciidae and Pocilloporidae), coral-associated symbiotic algae (Symbiodiniaceae), and a family of cyclopoid copepods (Xarifiidae) were shared between the two sampling approaches.

Using the ITS2 assay (designed to specifically target cnidarians and sponges), a total of 1,919,189 amplicon reads that passed QF were retrieved from 36 seawater samples (average per replicate = 54,644 ± 4,320 SEM or 56,206 ± 4,145 SEM excluding outlier sample DGX22) and 1,928,056 amplicon reads were retrieved from 39 butterflyfish gut samples (average per replicate = 50,335 ± 2,164 SEM; Appendix S4). In total, the ITS2 assay (combined gut contents and eDNA) produced eight eukaryotic phyla, 10 classes, 14 orders, 25 families, and 28 genera across the sample sites (Appendix S1; Fig. 2B). The average number of ZOTUs assigned per replicate of seawater was 21.28 ± 2.56 SEM and for gut content it was 27.05 ± 1.84 SEM. At sites where both seawater and gut contents were sampled, on average, 51.15 ± 17.69% SEM of the ZOTUs were unique to seawater, 27.43 ± 11.65% SEM of the ZOTUs were unique to gut contents, and 21.42 ± 8.85% SEM of the ZOTUs were shared between the two sampling approaches. Here, two families of stony corals were only found in gut contents (Agariciidae and Dendrophylliidae), whereas 16 families were unique to seawater, with the greatest representation included in the classes Anthozoa (sea anemones, stony corals and soft corals; six families) and Demospongiae (three families). Four families of stony coral were shared between the two sampling approaches, which included Acroporidae, Fungiidae, Merulinidae, and Poritidae. In many

A) 18S

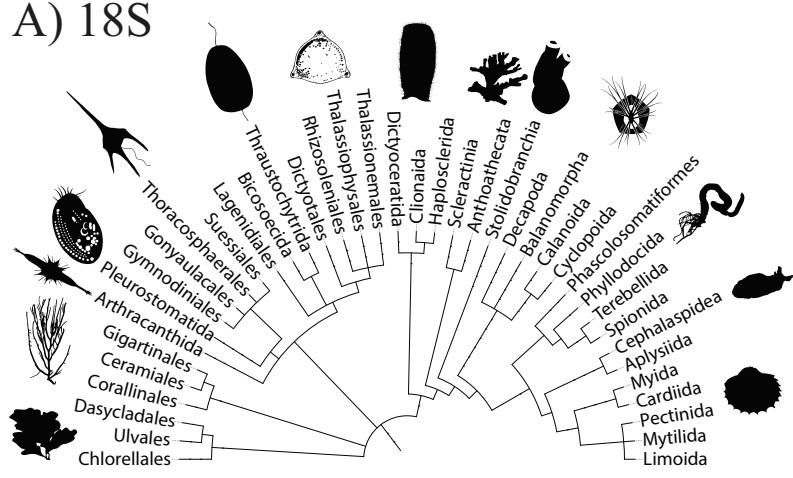

B) ITS2

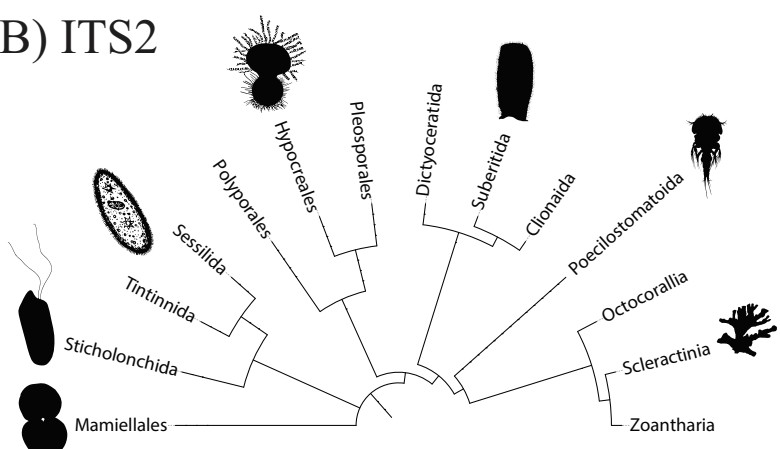

**Figure 2  Order level dendrogram of eukaryotic diversity detected at Dongsha Atoll (Taiwan) in the South China Sea based on seawater samples and oval butterflyfish (*Chaetodon lunulatus*) gut content samples.**                                           

cases, taxonomic assignment at the level of species was incomplete or ambiguous, reflecting gaps or uncertainty in reference material and/or taxonomy (8.3% and 9.5% identified to species for 18Suni and ITS2, respectively; Fig. 3).

## Seawater *vs* gut content metabarcoding

For 18Suni, we found that community assemblages identified in each sample based on ZOTUs was highly dependent on the approach (*i.e.*, seawater *vs* gut content; PERMANOVA: Pseudo-F = 43.426, df = 1, $p < 0.0001$; Appendix S2). This effect of sampling approach was consistent for the ITS2 assay (PERMANOVA: Pseudo-F = 11.089 df = 1, $p < 0.0001$; Appendix S2).

Based on PERMANOVA tests when the two sampling approaches were analysed separately, we found significant differences between sites for 18Suni sequences generated from seawater samples (Pseudo-F = 6.6691, df = 5, $p = 0.0001$; Fig. 4B) and gut content

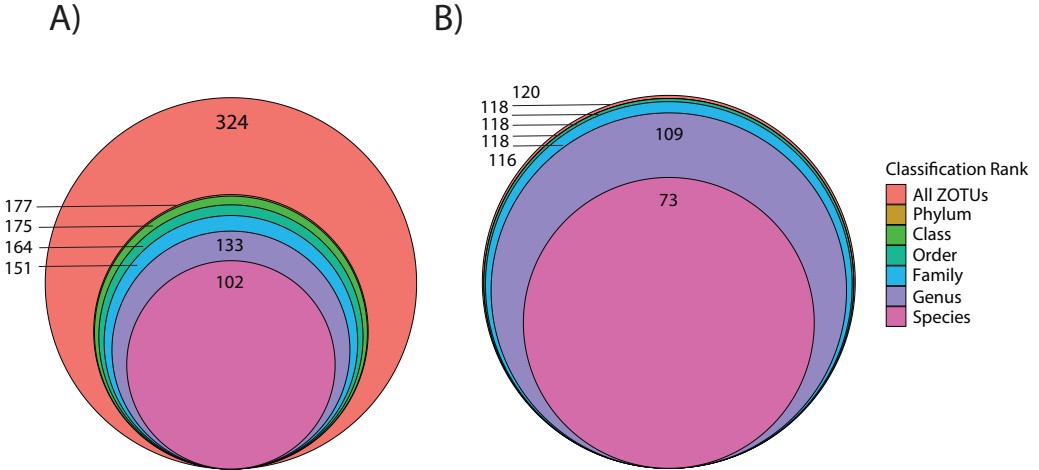

A)    B)

**Figure 3** Level of taxonomic resolution for ZOTU data from the 18Suni (A) or ITS2 (B) assays based on seawater samples and oval butterflyfish (*Chaetodon lunulatus*) gut content samples collected from Dongsha Atoll (Taiwan) in the South China Sea.

samples (Pseudo-F = 2.5081, df = 6, $p$ = 0.0001; Fig. 4A). Pairwise analyses however demonstrated that all sites were significantly different from each other for seawater samples (full results in Appendix S5), compared with only 33% of site comparisons being significantly different for gut content samples (full results in Appendix S5). Significant pairwise site differences for gut contents were found between 5 of the 7 sites: BG *vs* DS6, $p$ = 0.0261; BG *vs* DS10, $p$ = 0.0218; BG *vs* DS9, $p$ = 0.014; BG *vs* DS13, $p$ = 0.009; DS9 *vs* DS10, $p$ = 0.0025; DS9 *vs* DS13, $p$ = 0.0147; DS10 *vs* DS13, $p$ = 0.0013. DS4 and DWC were not distinguishable from other sites or each other based strictly on gut contents ($p$ > 0.05 in all cases) (Appendix S5). For 18Suni gut content samples, Pearson correlations (r > 0.7) indicated that site differences were driven by *Acropora* (ZOTU143, ZOTU553) and *Symbiodinium* (ZOTU696, ZOTU822) (Fig. 4A). In contrast, for 18Suni seawater samples, except for ZOTU168 (uncultured diatom) and ZOTU2487 (Thraustochytriaceae), all other ZOTUs driving site differences were identified as only Eukaryota (Fig. 4B).

 Based on PERMANOVA tests, we found significant differences between sites for ITS2 sequences generated from seawater samples (Pseudo-F = 3.5597, df = 5, $p$ = 0.0001; Fig. 4D) and gut content samples (Pseudo-F = 4.1188, df = 6, $p$ = 0.0001; Fig. 4C). Pairwise analyses indicated that all but one between site comparison (DS4 *vs* NSB) was significantly different for seawater samples. ITS2 sequences in gut samples appeared to be more distinct among sites compared to 18S sequences in gut samples, with 71% of site comparisons significantly different for the ITS2 assay (full results in Appendix S5). The specific pairwise ITS2 site differences between all sites for gut contents are provided in Appendix S5. Moreover, for ITS2 gut content samples, Pearson correlations were driven by corals in Fungiidae (ZOTU58, ZOTU106, ZOTU344) and *Montipora* (ZOTU53, ZOTU201, ZOTU262, ZOTU783) (Fig. 4C). For ITS2 seawater samples, Pearson correlations were driven by corals in Poritidae (ZOTU18, ZOTU44, ZOTU574) and *Montipora* (ZOTU153, ZOTU227, ZOTU240, ZOTU253, ZOTU337, ZOTU403, ZOTU472, ZOTU783) (Fig. 4D).

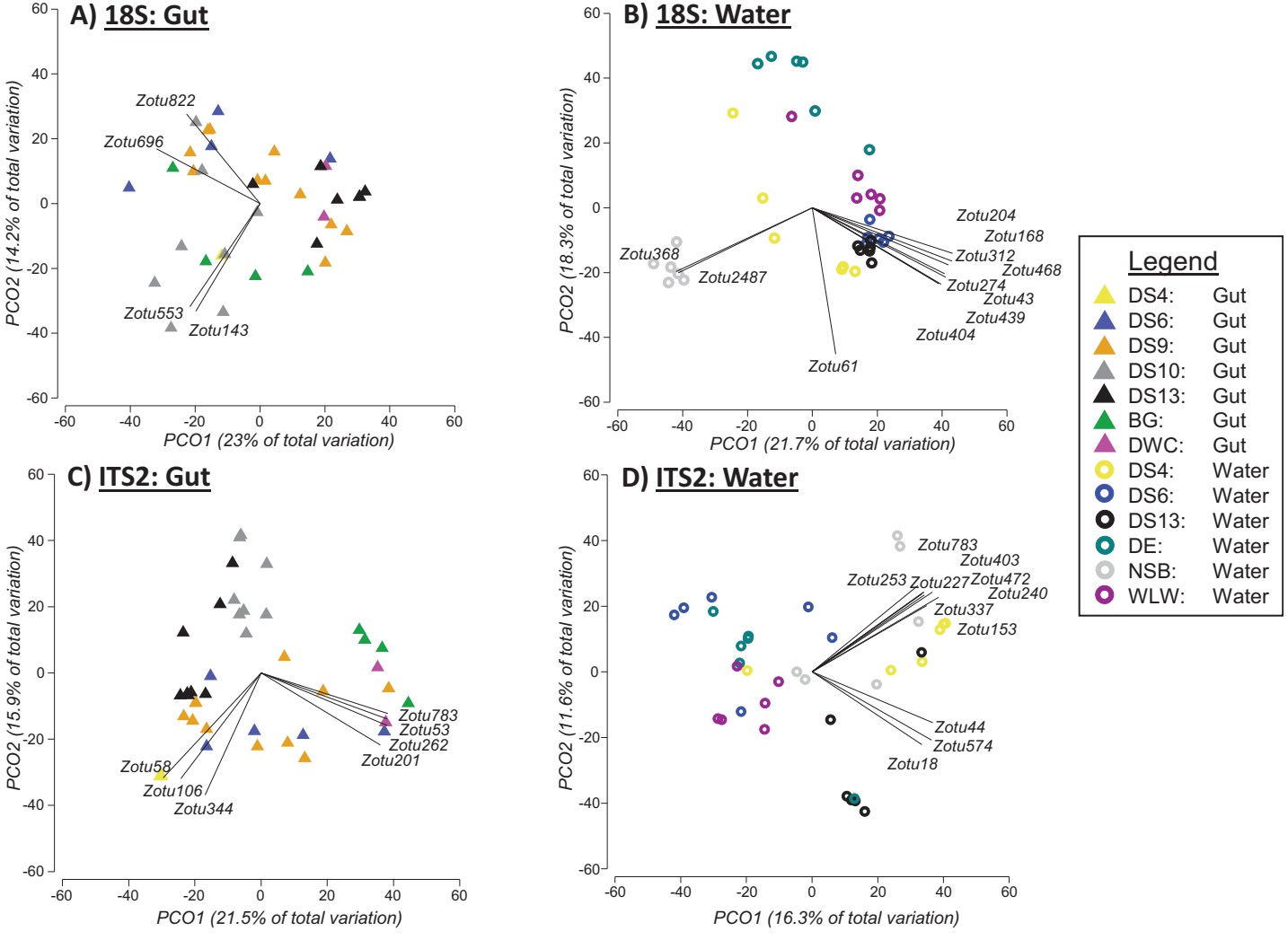

**Figure 4 Principal coordinate analysis (PCO) plots of presence/absence transformed ZOTU data (18Suni and ITS2 assays) based on seawater samples collected from six sites and oval butterflyfish (*Chaetodon lunulatus*) gut content samples collected from seven sites at Dongsha Atoll (Taiwan) in the South China Sea.** Characteristic ZOTUs are overlayed with Pearson correlations (r > 0.7).

To address the uneven sampling effort between seawater and gut content at individual sites, we repeated the above statistical comparisons, but this time including only those sample sites where both approaches were applied (DS4, DS6, DS13; Fig. 5). In this case, based on PERMANOVA tests with 18S sequence data, sampling approach (Pseudo-F = 23.986, df = 1, $p = 0.0001$; Fig. 5) and site (Pseudo-F = 2.6769, df = 2, $p = 0.0011$; Fig. 5) were significantly different. However, there was also a significant interaction between site and sampling approach (Pseudo-F = 2.889, df = 2, $p = 0.0006$; Fig. 5). For the ITS2 assay, both sampling approach (Pseudo-F = 6.7726, df = 1 $p = 0.0001$; Fig. 5) and site (Pseudo-F = 3.0838, df = 1, $p = 0.0001$; Fig. 5) were significantly different based on PERMANOVA, and again there was a significant interaction between site and sampling approach (Pseudo-F = 2.9675, df = 2, $p = 0.0001$; Fig. 5).

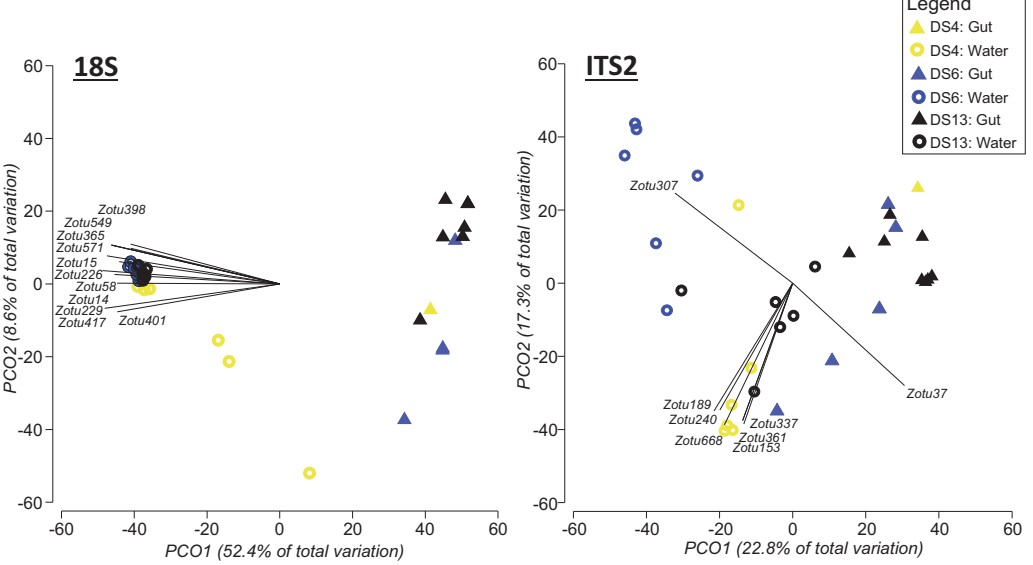

**Figure 5** **Principal coordinate analysis (PCO) plots of presence/absence transformed ZOTU data (18Suni and ITS2 assays) from seawater and oval butterflyfish (*Chaetodon lunulatus*) gut content samples collected in parallel at three sites in Dongsha Atoll (Taiwan) in the South China Sea.** Characteristic ZOTUs are overlayed with Pearson correlations (r > 0.8). Site abbreviations as follows: Dongsha Site 4 = DS4; Dongsha Site 6 = DS6; Dongsha Site 13 = DS13. In the case of 18Suni assay, except for ZOTU571 (*Paragymnodinium asymmetricum*), all other ZOTUs driving correlations were identified as only Eukaryota. In the case of the ITS2 assay, correlations were driven by *Porites* sp. (ZOTU37), *Dysidea* sp. (ZOTU307), and *Montipora* sp. (ZOTU153, ZOTU189, ZOTU240, ZOTU337, ZOTU361, ZOTU668).

Differences in semi-quantitative metrics for both assays such as relative read abundance, the percentage of positive replicates, and the number of ZOTUs among sites (Fig. 6) were consistent with differences detected among sites in community assemblages (Fig. 5). Moreover, families at individual sites with a high abundance of reads tended to have a higher percentage of positive replicates and a greater number of ZOTUs, indicating that these three metrics infer similar patterns. The most apparent difference within each assay, at least for gut contents, was the prevalence of stony coral reads (18Suni = 98.61% or 2,718,582 reads; ITS2 = 86.77% or 565,347 reads) *vs* non stony coral reads (18Suni = 1.39% or 38,326 reads; ITS2 = 13.23% or 86,220 reads); a difference consistent across all sampling sites. This pattern was less apparent for seawater, noting that the proportion of stony coral reads (18Suni = 0.23% or 3,465 reads ITS2 = 64.33% or 226,792 reads) to non stony coral reads (18Suni = 99.77% or 1,482,882 reads; ITS2 = 35.67% or 125,766 reads) was dependent on assay.

## DISCUSSION

In this study we used eDNA metabarcoding of seawater and gut content sourced from an obligate corallivore butterflyfish species resident to coral reefs in the South China Sea to test whether this species fed more broadly on benthic invertebrates as well as to compare biodiversity profiles inferred by the different sampling approaches. Based on 18S and ITS2 metabarcoding sequence data, we found that gut content detections were only partially

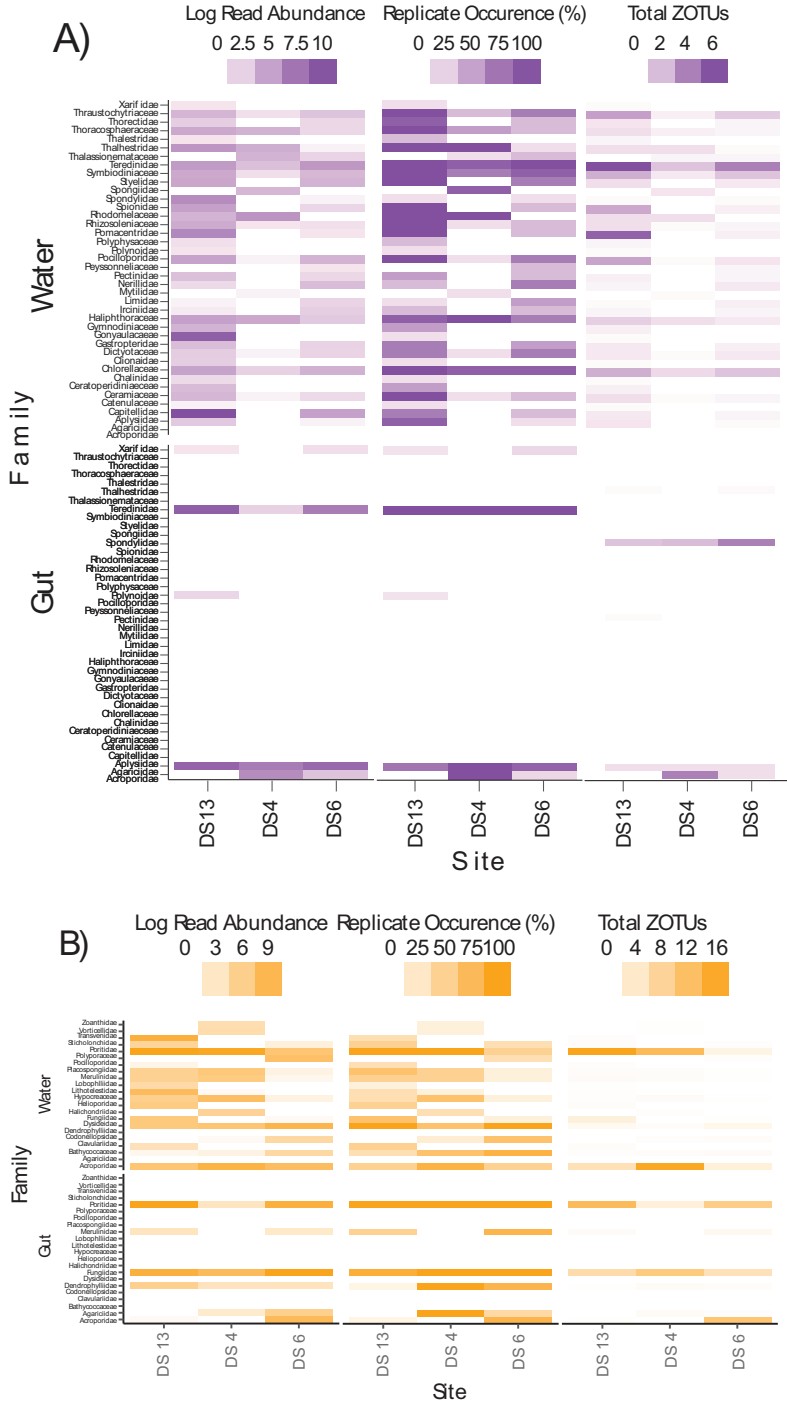

**Figure 6 Heatmap of log read abundance per site, percentage of positive replicates (*i.e.*, ZOTU presence) per site, and the total number of ZOTUs detected at the family level per site based on the (A) 18Suni and (B) ITS2 assay from seawater and gut content samples.** Note that only those sample sites where both methods were applied are included here, with site abbreviations as follows: Dongsha Site 13 = DS13; Dongsha Site 4 = DS4; Dongsha Site 6 = DS6. Log read abundance was applied due to the large-scale differences between ZOTUs with many reads *vs* few reads.

representative of the eukaryotic communities detected in the seawater samples, which is to be expected given that gut content is likely to be more selective than seawater. Key differences pointed towards gut content metabarcoding amplifying abundant benthic and encrusting dietary items that included, but were not exclusive to, hard corals, *vs* seawater metabarcoding amplifying a much broader spectrum of pelagic and benthic invertebrates. We hypothesise that some of the surrounding seawater and non-target taxa was likely ingested alongside prey targeted by the butterflyfish, thus providing a much broader feeding spectrum than expected. Whether these secondary food items provide a significant source of nutrients however will require further testing.

Although environmental data are available for the coastal ecosystems of Dongsha Atoll in the South China Sea, including sea surface temperature (SST; *DeCarlo et al., 2017a*), wave height (*Fu et al., 2012*; *Reid et al., 2019*), additional water parameters (*Chou et al., 2018*; *Reid et al., 2019*), and for some areas, live coral cover (*Dai, 2004*; *DeCarlo et al., 2017a*, *2017b*; *Tkachenko & Soong, 2017*), there are no comprehensive eukaryotic community composition data sets. To begin to remedy these deficiencies, we characterized the eukaryotic community composition in seawater samples to provide a snapshot of the overall diversity of the reef environment (Fig. 2). On the other hand, the eukaryotic community composition detected from fish gut content samples instead represented the breadth of targeted and, in some cases, non-target prey items consumed by the obligate corallivore that we considered. Comparing the eukaryotic community composition derived from both source materials and sampling approaches allowed us to characterize differences in their respective biodiversity profiles.

We found that for both the universal 18S assay and the more coral-focused ITS2 assay, fish gut contents were much more selective than the apparent availability of food in the surrounding environment as revealed by seawater metabarcoding. Based on the 18S sequences, the two major families in the gut contents of this putative selective feeder were stony corals from the families Acroporidae and Agariciidae, which contain species all commonly observed around Donghsa Atoll such as those in the genera *Montipora* and *Pavona* (*Cheng et al., 2020*; *Tkachenko & Soong, 2017*). In contrast, these same two families of stony corals were largely undetected in seawater samples, with detections instead dominated by unicellular chlorophytes, diatoms, and benthic invertebrates (*i.e.*, clams, worms). Although these latter groups include some of the most important animals in our oceans, with diatoms likely controlling up to 40–45% of oceanic primary production (*Mann, 1999*), it is clearly not important in the diet of this fish species, a result supported by *in situ* feeding observations of *C. lunulatus* at other coral reefs (*Pratchett, 2005*; *Pratchett, 2007*; *Nagelkerken et al., 2009*). Similarly, based on ITS2 sequences, stony corals were characteristic of the gut content samples, albeit a more extended set of families were detected with this assay (Acroporidae, Agariciidae, Dendrophylliidae, Fungiidae, Merulinidae, Poritidae). In addition to some of these coral families, seawater samples for ITS2 were characterized by multiple families of encrusting, boring, and calcareous sponges (Dysideidae and Placospongiidae, as examples). These results suggest that this butterflyfish may be selectively feeding on coral polyps, with modest incidental ingestion of other food sources, a result that is much clearer with the ITS2 assay that focuses on a smaller suite of

taxa. As applied by *Alexander et al. (2020)*, these ITS2 primers amplified a narrower range of taxa at a similar lagoon-dominated atoll in the Eastern Indian Ocean (*i.e.*, Cocos-Keeling Islands), although more coral taxa were recovered overall given that they collected more replicates per site ($N = 9$ *vs* $N = 6$), at more depths (3–5 and 10–13 m), and used two different assays targeting the ITS2 gene *vs* our single assay.

The number of available metabarcoding primers has increased significantly in the last decade (reviewed by *Takahashi et al. (2023)*), suggesting that a combination of assays is needed to maximise the diversity captured within environmental samples, regardless of the sampling approach. For example, assays based on general (*i.e.*, universal) primer sets are biased towards the more abundant DNA in a sample, whereas assays targeting a specific group or species are biased towards those groups or species. A combination of both types of assays should reveal a broad range of taxa, the specific groups or species targeted, in addition to sequences that cannot be assigned to taxa. Taxonomic uncertainty related to the latter is often allayed by the inclusion of detections sourced from traditional surveys (*e.g.*, *Kelly et al., 2017*).

The development of metabarcoding approaches to characterize corallivore diet over traditional observational methods is important because it enables the detection of digested soft-bodied taxa where morphological taxonomic expertise may not improve resolution. This genomic advance also allows for the refinement of future assessments of diet selectivity when paired with *in situ* surveys (*i.e.*, consumed taxa *vs* available taxa; *Pratchett, 2007*; *Lawton, Pratchett & Berumen, 2012*; also see *Ray et al., 2016*), particularly when DNA reference databases improve for invertebrate groups (*e.g.*, *Dugal et al., 2022*). One caveat to consider however is that gut content sequences may only reflect what the fish consumed in the past few hours given the relatively rapid gut clearance rates in corallivores and differential degradation rates of DNA (*Thomsen et al., 2012*).

The degree to which food items are passively ingested by butterflyfish *vs* actively selected by them is also an important consideration when trying to characterize the relative importance of detected prey in their guts (*Coker et al., 2023*). Indeed, detection of secondary ingestion is well documented in some systems (*Bowser, Diamond & Addison, 2013*; *de Bruyn et al., 2021*), particularly studies with other sources of information such as morphology and behavioural tendencies that allow them to make comparisons (*da Silva et al., 2019*). In our case, behavioural studies on the diet of this butterflyfish species (*Pratchett, 2005*; *Pratchett, 2007*; *Nagelkerken et al., 2009*), along with the specialised morphological structure of butterflyfish in general (*Goatley & Bellwood, 2009*), suggests that secondary ingestion may be negligible given the precise match between targeted corals in the field and corals detected in the lab based on our metabarcoding data. The results of our study indicate otherwise, with benthic, encrusting, and pelagic dietary items included alongside a predominantly hard coral diet for *C. lunulatus*. This finding is supported by gut content metabarcoding of a closely related obligate corallivore butterflyfish in the Red Sea, *Chaetodon austriacus*, which included the detection of non-Scleractinia DNA (*i.e.*, tunicates, worms, molluscs, algae, fungi, and sponges; *Coker et al., 2023*). In that same study, gut content metabarcoding revealed a similar taxonomic breadth of diet for the four other obligate corallivore butterflyfish species considered (*Coker et al., 2023*).

Symbiodiniaceae DNA was also detected in the guts of all butterflyfish considered in *Coker et al. (2023)* as well as *C. lunulatus* in our study, further suggesting that secondary ingestion may not be negligible given that many of these marine dinoflagellates would have been consumed alongside the coral polyps that they associate with.

As might be expected given the fine-scale spatial structuring of eDNA signatures in tropical marine environments (*e.g.*, *Alexander et al., 2020*; *West et al., 2020*), we found significant differences between the biota at all sites based on 18S and ITS2 data generated from seawater samples. Despite the heterogeneous distribution of corals and other reef inhabitants detected in the water column, the sequence composition of gut content samples was highly conserved. This is consistent with the strong dietary preferences of *C. lunulatus*, and the loss of condition and fitness associated with the consumption of suboptimal prey species such as *Porites* spp. and *Montipora* spp. (*Berumen, Pratchett & McCormick, 2005*). While *C. lunulatus* appeared to be selectively feeding, the fine-scale taxonomic resolution of the ITS2 marker identified important dietary differences between sites (Fig. 4C), which may be related to the availability of food sources owing to differences in coral assemblages across Dongsha Atoll.

Factors that may contribute to differences in biodiversity between methods and sites is variability introduced by the time of sampling, depth, season, habitat, as well as mixing from other coral reef locations. Indeed, each sample represents a snapshot in time of reef biodiversity and may not reflect biodiversity apparent at other times or associated with other habitats. For example, *Nichols, Timmers & Marko (2022)* found significant differences in OTU richness and abundance of marine eukaryotes based on the sampling of surface seawater when compared to seawater drawn directly from the crevices in the reef. In our study, this limitation is reflected in the more diverse seawater sample DNA when compared to the gut content sample DNA sourced from butterflyfish feeding on or near the reef matrix. As another example from temperate marine environments, *Jeunen et al. (2020)* found that vertical dispersion of eDNA from eukaryotic communities on the surface, *vs* samples sourced from 4 and 15 m depth, was limited. That said, we do not believe this limitation is universal. Indeed, substantial evidence based on field sampling that incorporates multiple survey techniques (*Jeunen et al., 2019*; *Stat et al., 2019*) and biophysical models (*Ellis et al., 2022*) show that biodiversity detections based on eDNA are highly localised, even in complex marine environments. In our study, we further attempted to control for the effects of time of sampling, season, and habitat by re-analysing our dataset based only on the three sample sites where both approaches were applied at the same time, which revealed similar patterns to the broader data set.

## Future directions

In this study, we explored the direct and indirect interactions of a corallivorous butterflyfish with their reef environment using two metabarcoding assays. An expanded treatment of gut content metabarcoding for fishes with different feeding preferences (*e.g.*, planktivores; *Leray et al., 2019*) or other resident animals with less selective requirements (*e.g.*, filter feeders; *Mariani et al., 2019*) could therefore be used to gather highly flexible datasets that address multiple ecological questions, particular in habitat that are difficult to

survey using visual approaches. For example, highly productive kelp forests and seagrass meadows contain an abundance of organisms that are often small, epiphytic, symbiotic, or cryptic. If interested in a narrower range of taxa corresponding to the food spectrum, metabarcoding assays of the surrounding seawater (*Port et al., 2016*; *Stat et al., 2019*; *Lamy et al., 2021*; *Momota, Hosokawa & Komuro, 2022*) in tandem with gut contents of the animals that feed directly on them (*Gajdzik et al., 2021*; *Díaz-Abad et al., 2022*) can prove to be a more robust approach to validate the resulting detections. Moreover, given the sensitivities of DNA-based sampling approaches, we suspect that in the future, with further improvements to taxonomic assignment or a renewed focus on microbiomes, gut content metabarcoding may be able to estimate fish fitness as it relates to changing environmental conditions and/or habitat degradation (*Clever et al., 2022*).

## ACKNOWLEDGEMENTS

The authors would like to acknowledge TrEnD staff, notably Matthew Power and Megan Coghlan, for DNA sequencing assistance. We would also like to thank Mahsa Mousavi-Mousaviderazmahalleh and Georgia Nester for bioinformatic assistance. At Dongsha Atoll research station, we thank Dr Yalan Chou, as well as members of the MISE Laboratory at the University of the Ryukyus for their assistance in the field.

**Data archiving statement**

The demultiplexed FASTQ sequences for each sample are accessible *via* GenBank accession numbers SAMN35301691 to SAMN35301726 (18S seawater), SAMN35301738 to SAMN35301773 (ITS2 seawater), SAMN35301166 to SAMN35301203 (18S gut content), and SAMN35301492 to SAMN35301530 (ITS2 gut content) at the following link: https://www.ncbi.nlm.nih.gov/bioproject/PRJNA974888.

These same files are also accessible *via* Dryad Digital Repository accession number https://doi.org/10.5061/dryad.dbrv15f5x at the following link: https://datadryad.org/stash/share/ykBnyetly-OnOOf2qtfHxPPSfWZWgzL_2qZpe6TgFMI.

### Funding

This study was funded by the Australian Research Council Linkage Projects (LP160100839 and LP160101508) to Joseph D. DiBattista and Michael Bunce, a Dongsha Atoll Research Award under funding (108-2119-M-110-005) issued by the Ministry of Science and Technology to Joseph D. DiBattista and Shang-Yin Vanson Liu, as well as a Curtin University Early Career Research Fellowship to Joseph D. DiBattista. We also acknowledge support from the Pawsey Supercomputing Centre with funding from the Australian Government and the Government of Western Australia. The funders had no role in study design, data collection and analysis, decision to publish, or preparation of the manuscript.

### Grant Disclosures

The following grant information was disclosed by the authors:
Australian Research Council Linkage Projects: LP160100839 and LP160101508.

Dongsha Atoll Research Award under funding: 108-2119-M-110-005 issued by the Ministry of Science and Technology.
Curtin University Early Career Research Fellowship.
Australian Government and the Government of Western Australia.

## Competing Interests

The authors declare that they have no competing interests.

## Author Contributions

- Joseph D. DiBattista conceived and designed the experiments, performed the experiments, analyzed the data, prepared figures and/or tables, authored or reviewed drafts of the article, and approved the final draft.
- Shang Yin Vanson Liu conceived and designed the experiments, performed the experiments, analyzed the data, prepared figures and/or tables, authored or reviewed drafts of the article, and approved the final draft.
- Maarten De Brauwer analyzed the data, prepared figures and/or tables, authored or reviewed drafts of the article, and approved the final draft.
- Shaun P. Wilkinson analyzed the data, prepared figures and/or tables, authored or reviewed drafts of the article, and approved the final draft.
- Katrina West performed the experiments, analyzed the data, prepared figures and/or tables, authored or reviewed drafts of the article, and approved the final draft.
- Adam Koziol performed the experiments, analyzed the data, prepared figures and/or tables, authored or reviewed drafts of the article, and approved the final draft.
- Michael Bunce conceived and designed the experiments, authored or reviewed drafts of the article, and approved the final draft.

## Animal Ethics

The following information was supplied relating to ethical approvals (*i.e.*, approving body and any reference numbers):

Fish collections were undertaken in accordance with the policies and procedures drafted by the Animal Care and Use Committee of National Sun Yat-Sen University. Most fish were dispatched by the action of the spear, but those that were not, were subject to immediate ike jime and placed in an ice slurry on board the vessel. Field permissions to undertake the research were obtained from the Ministry of Interior in Taiwan under permit No. 1070001035.

## DNA Deposition

The following information was supplied regarding the deposition of DNA sequences:

The demultiplexed FASTQ sequences for each sample are available at GenBank: SAMN35301691 to SAMN35301726 (18S seawater), SAMN35301738 to SAMN35301773 (ITS2 seawater), SAMN35301166 to SAMN35301203 (18S gut content), and SAMN35301492 to SAMN35301530 (ITS2 gut content).

The same demultiplexed FASTQ sequences for each sample are also available at Dryad: DiBattista, Joseph et al. (2023). Gut content metabarcoding of specialized feeders is not a

replacement for environmental DNA assays of their reef environment [Dataset]. Dryad. https://doi.org/10.5061/dryad.dbrv15f5x.

## Data Availability

The demultiplexed FASTQ sequences for each sample are available at Dryad: DiBattista, Joseph et al. (2023). Gut content metabarcoding of specialized feeders is not a replacement for environmental DNA assays of their reef environment [Dataset]. Dryad. https://doi.org/10.5061/dryad.dbrv15f5x.

## Supplemental Information

Supplemental information for this article can be found online at http://dx.doi.org/10.7717/peerj.16075#supplemental-information.

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
