# Peer review of "Gut content metabarcoding of specialized feeders is not a replacement for environmental DNA assays of seawater in reef environments"

_PeerJ, doi:10.7717/peerj.16075_

## Round 0.1 · original submission · Major Revisions

As you can see the reviewers have provided a thorough but overall positive critique on your paper. I am looking forward for your revised version along with the point-by-point reply to all comments.

·

Basic reporting

Overall, this paper is written well in a clear and concise manner, which is easy to understand for readers from any field. The authors make efficient and proper use of relevant literature and provide sufficient background. The article structure follows the journal's format and raw data is accessible. The figures make sense with the content of the paper and help to support the authors' claims. I would suggest minor improvements for the following figures:

Figure 1. I would highly suggest including in this figure a smaller map to show the location in the South China Sea so that the reader can understand where this atoll is in relation to its greater geographic location (Taiwan).

Figure 2. All the text (numbers and classifications) needs to be much larger, in its present state it is challenging to read.

Figure 3. I would try to increase the text size if manageable inside and outside the PCO plots. Same with the supplemental figure PCO.

An additional comment is that I felt that the title was a little misleading with the mention of 'environmental DNA assays', as seawater was the only assay of the reef environment, when I initially read it I thought multiple types of eDNA assays were conducted. I would suggest the authors add the word 'seawater' in the title to make it more clear.

Experimental design

The authors mention that the sampling period occurs only over one week, and for seawater, which is highly variable, this is only a snapshot in time of the reef biodiversity that could be in the area, especially when seasonal changes heavily impact the type of diversity one may acquire in their seawater sampling. The variation in how many sites were sampled, different areas of the reef, etc. are good to try to account for changes in diversity, but I think a solid explanation and discussion surrounding this variability that could occur in their results in relation to gut content diversity is needed.

In the materials and methods, the authors state how many replicates of water were collected, but the number of biomass samples from butterfly fish gut contents was not mentioned until way later. It is necessary to mention this at the beginning of the methods.

Line 138 indicates that water was collected adjacent to habitats at only 30 cm below the surface (it is not clear whether it was 2 meters adjacent at sites underwater and if so what depth? or if it was 2 meters next to the site at 30 cm depth), but the depth could play a HUGE factor in changes between ambient seawater and seawater collected directly at the reef habitat where butterfly fish occupy and between crevices of the reef. Nichols et al. 2021 (https://onlinelibrary.wiley.com/doi/full/10.1002/edn3.203) mention this very effect of sampling in ambient seawater vs. directly at the reef: “Based on the data from our four survey methods, patterns of MOTU richness and abundance of DNA reads suggest that broad eDNA surveys from ambient seawater do not fully capture reef cryptobiome diversity. Ambient and crevice communities only shared 5% of total MOTUs and were significantly distinct from each other.” This ties into my other comments regarding my concerns about being aware of the context in which the authors are comparing ambient seawater collected just 30 cm below the surface, directly to the gut contents of fish that are feeding in specialized habitats directly on the reef. I think the authors need to heavily address the reasoning for this and reframe the context of comparing seawater almost at the surface of the reef, which is subject to currents bringing in DNA from several other areas to the gut contents of feeders directly on the reef.

Line 145 - Mention how many times the water was filtered for each 1 L replicate as the filtration amount and water volume are known to change results, therefore it is important to mention.

Validity of the findings

There is still a lot of work that the authors need in their discussion to reframe the context of this experiment. There are a lot of considerations that were not fully explained that leaves a lot of questions. Line 391 mentions that this 'provides a snapshot of the overall diversity of the reef environment', but it is not entirely an 'overall' method for the reef diversity at these sites, as this is dependent on season, depth of sampling, type of sampling, etc. I would look at it as a thermometer reading of diversity from a specialized feeder and the reef environment in which it is feeding.

Additionally, the biodiversity identified from seawater samples (if only collected at 30 cm depth) is more of a diversity in the region in which all of the sites were located. The authors do not sufficiently address these factors to explain why they may have received the results that they did. Sampling seawater over 7 days (which can also differ depending on flow regime and currents) at a shallow surface depth only provides a snapshot of diversity for those particular locations in that April/May spring season, but not during other seasons throughout the whole year, which would provide an 'overall biodiversity snapshot'. The sampling time and effort are not the issues, but more so I would suggest the authors be a bit more careful with how they phrase their results in the bigger picture of what they are showing.

The authors need to address these other points as well:

The results make sense as to why seawater was much more diverse and abundant, especially when you see the butterfly fish are only feeding on a few species of corals lines 399-400 and also in Figure 4 in the heatmap -> makes me think that the water is picking up DNA from other areas - this should be addressed by authors.

Additional comments

There were a few other minor things to fix:

“Lagoon limited in open water exchange” line 115 – This was mentioned but is this an explanation for why the seawater collected is a good representation of the reef habitat? If so, this point needs to be explained more, and how the authors know this in the discussion.
Lines 124-126 are not super clear – need to rewrite to better express the point of the connection between varying habitat types and the health of reefs.

General comments:
Authors have to keep in mind that seawater is highly variable and although a good indicator of ecosystem diversity, should also remember that seawater can come from different regions in the area and may explain the high variability and differences in gut contents. This point needs to be addressed in the discussion section. There is definitely a story here, but the manuscript needs to be reframed to support a stronger narrative. What I feel that I am missing is a strong case of importance to understand the main question, regarding the differences between these two sampling methods as a snapshot of the biodiversity of the reef. I almost feel that they are comparing two different things. One is a snapshot of the specific biodiversity at a reef from a specialized feeder, while the other is from ambient biodiversity in the region of the reef. There needs to be a better explanation to tie the reasoning behind these approaches together and address these issues to bridge the gap between the methods and the differences that the results showed. This will help to frame the idea behind combining approaches (and not just as stand-alone methods) of eDNA sampling methods to obtain a bigger picture idea of reef biodiversity and health. I also suggest that the authors explain a broader connection of the importance of this work, how it should be applied to their future work, how practitioners can adopt this, and how it is important to have this information/relevance to coral reef biodiversity surveys.

·

Basic reporting

The study "Gut content metabarcoding of specialized feeders is not a replacement for environmental DNA assays of their reef environment" by DiBattista et al. is an interesting read. They compare a multi-assay metabarcoding approach of environmental DNA (eDNA) from seawater and partially digested intestinal parts of an obligate coral feeder (Chaetodon lunulatus) at coral reef sites in Dongsha Atoll in the South China Sea. I commend the authors for their extensive data and detailed statistical analysis. The manuscript is clearly written and uses professional, unambiguous language. The literature is referenced where appropriate and sufficient background and context is provided. All appropriate raw data are provided and checked for availability by me. While supplementary material is referenced in the text, I could not find any reference to the deposit of raw sequencing data in the manuscript. However, there is a statement of data availability in the online version, but this would need to be mentioned in the final manuscript. I also have a few comments regarding some ambiguities or where more information is needed. Also, some of the figures and supplementary material need a bit of revision to provide more information for future readers. I will go into this in detail in the general comments.

Experimental design

The manuscript is original primary research that is within the goals and scope of the journal. However, the research question could be better defined. The strength of this study is that it demonstrates the utility of combining metabarcoding assays to study the diet of a fish while simultaneously characterizing the habitat. However, the focus of the work of the gut contents of the butterflyfish species Chaetodon lunulatus, an obligate hard coral feeder, is laid on the fact if it can be used to characterize the complete eukaryotic diversity of the habitat. While reference is made to studies (Coker et al. 2023 and de Bruyn et al. 2021), in which the authors are partly involved, that indicate secondary signaling and involuntary uptake of DNA during feeding, these studies do not provide evidence that this method can be extended to the full eukaryotic diversity, especially in a fish that feeds as selectively as the species in this study. For example, Coker et al. 2023 used a COI primer (Leray et al. 2013) that is also used in eDNA studies of the entire eukaryotic diversity of a habitat, but the results from the stomach contents of the butterfly fishes studied appear to be quite specific to the preferred diet. The paper would benefit from a revision that focuses on the diet of the butterflyfish species, Chaetodon lunulatus, and characterizes the composition of the benthos near the butterflyfish feeding sites using seawater samples. This is also mentioned in the introduction, but not rigorously followed up in the manuscript. The methods are described in detail and comprehensively, with a few minor exceptions (see General Comments). The analyses are valid and correspond to common methods in the analysis of eDNA datasets and their statistical analysis.

Validity of the findings

The raw sequencing data are provided in a discipline-specific repository, while the results of the eDNA analysis and the tables used in the statistical analysis are provided as supplementary material. Although the conclusions are well formulated and linked to the original research question, the manuscript could benefit from a shift in focus, as mentioned earlier, which would require minor revisions to some graphs and tables, which I discuss in detail in the general comments.

Additional comments

In the following, I will individually address issues that need revision or clarification and cite the corresponding lines in the manuscript.
Introduction:
Lines 103-109: This section is a good reflection of what the study should focus on and what the strength of this study is and what can be well demonstrated with the methods and analyses used here.
Line 114-115: I don`t know if it`s just the format but there seem to be some extra spaces in here.
Line 121-123: Does this sentence refer to all corals on the atoll or just the lagoon corals. Can you rephrase the sentence to clear that up?
Line 127-131: This is the second section in the introduction that discusses the goals of the paper, but it is different from the first (see above, Line 103-109). I think the introduction could use a rewrite at this point by combining the two sections and focusing on the goals in the first. Obviously, the hypothesis of testing the eukaryotic community from gut contents can still be part of the paper. However, I would not focus on this, since the previous studies already had enough evidence that this is probably not promising, as is also shown by the results.
Material and Methods
Line 136-138: Can you elaborate more how the water samples were taken, snorkeling, diving, from the boat? How big were the Nalgene bottles? Approximately how high above the reef/seagrass/ground were you during sampling?
Line 139-140: I don`t understand this sentence completely, as I don`t know to what "this statistical factor" refers to. Also, I don`t see this information in S4 and if S1 is meant here I see that for later statistical analysis, some samples are treated as pairs because seawater and gut content was both collected at some sites. Please rephrase this sentence to make clear what is meant here and also look that the right supplement is mentioned here.
Line 141-144: Can you add more information on how long it takes to filter a single sample, and how much time elapses between the first and the last sample of a filtering process?
Line 166-170: Why was it not possible to collect water samples after catching the fish, to make sure you have enough? For some sites you collected quite a lot of fish, for one site you only have one fish that goes into analysis.
Line 173-177: Please add how you treated the filters. Were there lysed as a whole or were they subjected to some kind of shredding.
Line 187: I think you mean the gene here and not the rRNA itself.
Line 236: Just a small thing. For USEARCH you put the version number in parentheses while for other programs not. I would make it consistent throughout the manuscript.
Line 263: ZOTU1841 is still present in the ZOTUSclean sheet in the Appendix. I see it`s removed for the PRIMER analysis but maybe already remove here or mark it that this ZOTU is not used any further, also in the sheet.
Line 286: Was the initial PERMANOVA was also done in PRIMER v 7 and based on presence/absence format?
Results:
Line 304-305: Is the average number of ZOTUSs including DGX22 that had 3 reads according to S4 and only 1 read left in the ZOTU file in S1? Why inlcude it?
Line 306-308: Remove sentence or put into discussion.
Line 308: Instead of metabarcoding assay, maybe better write ITS2 assay, as it could otherwise sound like the complete assay of 18S and ITS2.
Line 352-353: As seagrass was only sampled for seawater you probably mean the overlap in the PCo. It’s confusing at this point as the sentence before and after referring to gut content. Maybe move the sentence to another location. Also, you said before that almost all comparisons for locations were different in the pairwise analysis, so you should make clear what you mean here.
Line 366-371: You could maybe add how high the proportion of stony coral reads vs. non coral reads in your data for 18S and ITS2 is in the gut contents between the different sites were. I think something along the lines of Fig. 2 in Coker et al. 2023 or at least a report on the numbers, because so far there is no actual summary of what the diet of the fish now actually is. I think this would allow the reader to better understand how well the metabarcoding worked to determine the dietary preference of the butterflyfish.
Discussion
Line 376-378: For both primers you looked at the corresponding genes, not the rRNA.
Line 378-382: Preferentially amplifying sounds like a primer bias or the metabarcoding assay selecting for certain taxa but the thing here is that the approach can only amplify stuff that is actually in the gut. Please rephrase the sentence to avoid any confusion of what you intend to say here.
Line 389-391: It would be great if the composition of the eukaryotic community could be graphically depicted in some way in the paper, as you mention there is nothing like this for Dongsha Atoll yet, perhaps similar to Figure 2 in West et al. 2020 Also, if there is data on coral cover, it would be worth mentioning here if it matches your data. Have you found something that has not been described before for this site? How much of the coral diversity described for this atoll is found in your data? I think this is important to validate the robustness of your approach and also important data to have for the future.
Line 394-395: You have described statistical differences in the biodiversity profiles and there are exclusive ZOTUs named in the result for each of the assays and sampling methods. The audience would benefit from a graphical representation. Perhaps you could revise Figure 5 to include not only the total ZOTU representation but also gut and seawater. This would give a direct indication of which ZOTUs are included in the different sampling methods and where the differences are.
Line 415-420: Do you have rarefaction curves of your samples based on your sampling locations and method? I mean for one of the fish samples you only have one fish so it is clear that you did not get the full diversity of the site, but how does it look for other sites? Maybe you can add here that there are potentially other marker systems to look at coral diversity using eDNA and reference a few.
Line 427-438: You could also reference Coker et al. 2023 and expand a little bit on secondary and involuntary digestion. Was there evidence in those studies and to what extent? Were you expecting a lot of signals other than prey organisms of the fish? For example, for 18S you have some copepods in the gut data. Do you think these could be prey of scleractinian corals or are they just accidentally ingested? Do the Symbiodiniaceae also contribute to the fish diet or are they just accidentally ingested with the corals?
Line 446-449: There is no 3c, if you are referring to the third graph in Figure 3, please add letters to the figure accordingly. In addition, there are differences for gut and seawater community composition in your results. Are they the same? Can you add a little bit about how these sites actually differed in coral composition between gut contents and seawater contents?
Line 452-463: You could also expand on the possibility of combining metabarcoding of the fish gut content with the measurement of certain fish health parameters. You report differences in gut composition between sampling sites, so there could be differences in the fitness of these fish, with implications for changing environments under climate change.
Line 459-463: You should add that your results suggest this if you are interested in a narrower range of taxa corresponding to the food spectrum, but not necessarily to get a broader overview, given the difference in the number of ZOTUs for 18Suni between seawater and gut contents.
Conclusion
Line 466-474: The conclusion should be rewritten so that it is consistent with the proposed change in the focus of the work.
Figures
Figure 2: The first legend entry should probably be all ZOTUs and not all sequences? As mentioned above, it would be good to include some kind of overview of the eukaryotic community of the Dongsha atoll here.
Figure 3: Maybe add letters to the graphs when you refer to them in the text. Would also help with the figure legends and what ZOTU is being referred to at what time.
Figure 5: As mentioned above, perhaps you could revise Figure 5 to include not only the total ZOTU representation but also gut and seawater. This would give a direct indication of which ZOTUs are included in the different sampling methods and where the differences are.
Supplement
S1: Could you make additional sheets for 18S and ITS2 that only include ZOTUs for seawater, respectively gut content. At the moment it is difficult to understand which ZOTUs appear for the respective collection methods.
S2: The order of the legends is not consistent between the different plots. On some it’s sorted for location while on other it’s sorted for sampling method/substrate
S5: The p-value for the comparison of DS4 vs. NSB is printed in bold, despite being not significant. Can you add in the legend why you need to do a Monte Carlo test?

·

Basic reporting

The article written in English and clear, unambiguous, technically correct text.
The structure of the article meet the format of ‘standard sections’.
Figures relevant to the content of the article, of sufficient resolution, and appropriately described and labeled.
All appropriate raw data have been made available in accordance with Data Sharing policy.

Experimental design

150-152 Please provide number of fish samples and the size of fish.

The investigation conducted rigorously and to a high technical standard. The research must conducted in conformity with the prevailing ethical standards in the field.

Validity of the findings

The data robust, statistically sound, and controlled.

The conclusions appropriately stated, connected to the original question investigated, and limited to those supported by the results.

Additional comments

Lines 52-54 should be more detailed to provide clarity about the results of the study.

---

## Round 0.2 · Minor Revisions

There is one minor editorial issue that needs to be fixed. Please provide us with the map as suggested by the reviewer so we can proceed to the final decision.

·

Basic reporting

The authors did a great job correcting and improving upon revisions, overall nice work.

Experimental design

.

Validity of the findings

.

Additional comments

.

·

Basic reporting

Basic reporting
I have looked through the manuscript and the authors' responses to my comments and I am satisfied and agree with the revised version.

Experimental design

Experimental design
I have looked through the manuscript and the authors' responses to my comments and I am satisfied and agree with the revised version.

Validity of the findings

Validity of the findings
I have looked through the manuscript and the authors' responses to my comments and I am satisfied and agree with the revised version.

Additional comments

Additional comments
I have just one small comment. I'm on the other reviewer's side of adding an overview map to figure 1, however now the satellite picture of Dongsha Atoll with the individual collection points is really small and the font hard to make out. I think it would be good to make the satellite image larger and embed the South China Sea location smaller for it, if possible. Otherwise, I am satisfied with the changes that have been made.

---

## Round 0.3 · accepted · Accept

Thank you for your patience and detailed revision. Your paper can now be accepted.